Open access Original research

# Coproducing a physical activity referral scheme in Germany: a qualitative analysis of stakeholder experiences

Sarah Klamroth ![ORCID],[1] Eriselda Mino ![ORCID],[1] Inga Naber,[1] Anja Weissenfels ![ORCID],[1] Wolfgang Geidl,[1] Peter Gelius,[2] Karim Abu-Omar,[1] Klaus Pfeifer[1]

¹Department of Sport Science and Sport, Friedrich-Alexander-Universität Erlangen-Nürnberg, Erlangen, Germany
²Insitute of Sport Sciences, Université de Lausanne, Lausanne, Switzerland

**Correspondence to**
Dr Sarah Klamroth;
sarah.klamroth@fau.de

## ABSTRACT

**Objectives** This study evaluated stakeholders' experiences of participating in a coproduction process to develop a physical activity referral scheme (PARS) in the German healthcare system. The focus was on examining facilitators and challenges, along with gathering insights on potential modifications to the joint development process, all from the viewpoint of stakeholders.

**Design** This qualitative study employed one-to-one semi-structured interviews, and the findings were analysed using summarising qualitative content analysis.

**Setting** The study focused on the German healthcare system.

**Participants** Seven stakeholders from the coproduction process were purposefully selected for interviews using maximum variation sampling. The interviewees represented different sectors (physician associations, physical activity professionals' associations, health insurance companies and patient organisations), various positions within their organisations, and different levels of attendance during the coproduction process.

**Results** In almost all interviews, the following factors were highlighted as facilitators of the development process: coproduction approach, process of coproduction, multi-sector stakeholder group, possibility of active participation, coordinating role of researchers, communication, atmosphere and interaction. In contrast, differences in roles and hierarchy, merging of different perspectives, clarification of intervention costs, and competition and conflicting interests were pointed out as challenges. Only a few suggestions regarding adaptations in terms of group composition and cooperation among stakeholders were mentioned.

**Conclusions** Stakeholder experiences with the joint development process were predominantly positive, indicating that coproduction is a beneficial approach for the development of PARS intended for integration into healthcare systems. The effective management of power differences among stakeholders is intricately tied to the coproduction method; therefore, it should be selected carefully. The research team plays a pivotal role in coordinating and negotiating the process, and the team should be equipped with a diverse set of skills and knowledge, particularly to understand the intricacies of the

## STRENGTHS AND LIMITATIONS OF THIS STUDY

⇒ This is one of the first studies evaluating a coproduction approach for the development of a physical activity referral scheme (PARS).

⇒ The in-depth qualitative analysis of interviews effectively brought to light facilitators, challenges and potential adaptations in the joint development process from a stakeholder perspective.

⇒ Because of a time lag between the coproduction process and the interview sessions, there is a possibility that memory biases may have influenced the stakeholders' responses.

⇒ Although this study appears to be primarily relevant to the German healthcare system, it also provides relevant information for using of coproduction processes to develop and implement innovative forms of care (such as PARS) in other healthcare systems.

healthcare system where the PARS is intended for implementation.
**Trial registration number** NCT04947787.

## INTRODUCTION

The importance of promoting physical activity (PA) in healthcare settings is widely recognised. PA not only enhances the health and well-being of healthcare service users but also holds the potential to alleviate the economic burden on public healthcare systems.[1] Conversely, healthcare systems, given their expansive reach, can play a pivotal role in achieving global targets for reducing high levels of physical inactivity.[2 3] Physical activity referral schemes (PARS), in which healthcare professionals provide PA-related advice and refer patients to specialised PA services or programmes, have been recommended as a promising strategy to achieve the aforementioned objectives.[4 5]

Despite extensive research and occasional national-level implementations of PARS,[6] the evidence regarding their effectiveness remains inconclusive.[7 8] This ambiguity may stem from substantial variations in scheme design and content.[9 10] Another potential

contributing factor is the oversight of healthcare system features as an important component of intervention complexity.[11] Coproduction approaches, increasingly popular in health services research,[12] can be a valuable instrument in addressing this overlooked aspect in intervention development.[13] While lacking a universally accepted definition,[14] in the context of this paper, coproduction refers to the participation of key stakeholders in the intervention development process. Only a few PARS have used coproduction as a design tool,[15 16] and there have already been appeals to enhance its utilisation.[17]

The integration of health services, such as PARS, into the healthcare system requires evidence of effectiveness, encompassing both the scheme itself and its implementation.[18] However, due to the significant variability in PARS,[9 10] there is little guidance on developing an effective model tailored to a specific healthcare context. Involving stakeholders, such as service users, healthcare providers, relevant organisations and policymakers, in the design process may aid in navigating the international ambiguous evidence base and reassessing it against the national health system characteristics. Moreover, involving stakeholders relevant to the system can contribute to understanding the context and prioritising and fostering productive problem-solving.[19] Thus, the knowledge and experience of these stakeholders, who are critical for the implementation process, provide valuable information for the implementation plan.

Involving relevant stakeholders in the intervention development process builds a sense of ownership, establishing a connection that encourages people to identify with the project and become invested.[20] This, in turn, can enhance the feasibility of the new PARS, as healthcare professionals are more likely to accept their new roles and associated responsibilities. This aligns with what Greenhalgh et al[21] call 'assimilation by the system'—a process necessitated when there is a need for change in existing structures and workflows. It is assumed that individuals involved in the design process are more inclined to embrace the 'winds of change', in contrast to situations where a 'product' is imposed on them (eg, a top-down approach to policy development), and they react with reluctance to change.[20]

Against this backdrop, the *BewegtVersorgt* project employed a coproduction approach with the aim of developing, implementing and evaluating a PARS model tailored for individuals with non-communicable diseases (NCD) in the German healthcare system.[22] This system is complex and fragmented in its organisation, service provision and financing.[23] As such, it encourages a rather gradual approach to change, which makes transformation at the system level challenging and slow-paced. PA promotion is also affected by this fragmentation and incrementalisation, leading to uncoordinated efforts and stagnation of PA promotion in healthcare settings.[24 25] Given this, the design and implementation of a PARS using a top-down approach or adopting an international model with demonstrated effectiveness would yield questionable results in the German context. Moreover, much of the financial governance is vested in corporatist institutions like health insurance companies,[23] which hold decision-making power concerning the reimbursement of new health services, such as a PARS. To effectively navigate the complexity and rigidity of the German healthcare system and optimise conditions for the incorporation of the PARS, it is imperative to involve key stakeholders from policy and practice in the PARS development process.

Despite the perceived potential of coproduction approaches in health services research, they have rarely been evaluated.[17 26] Reflecting on this process may help identify areas of improvement for more efficient and productive coproduction efforts or facilitate decision-making regarding its appropriateness for future projects. In a previous commentary, we reported the experiences with coproduction from researchers' perspective.[25] In this paper, we complement the landscape with the evaluation of the coproduction process from the stakeholders' point of view. The following are the specific subobjectives:

▶ To identify facilitators and challenges from the perspective of stakeholders participating in a coproduction process for the development of the PARS.
▶ To collect stakeholders' suggestions for adapting the coproduction process with the aim of enhancing it for future use.

## METHODS
### Study design
The *BewegtVersorgt* project is designed to encompass the development, implementation, evaluation and scaling-up of a PARS for individuals with NCD. The project unfolds in four distinct phases. In the initial two phases (June 2019 to November 2021), the research team collaborates with relevant stakeholders in a coproduction process to develop a PARS model. Specifically, the second phase involves preparing for the implementation of the developed PARS as a model project in routine care. The third phase (December 2021 to December 2023) focuses on the effectiveness and implementation success of the coproduced PARS. This evaluation is being carried out in the Nuremberg Metropolitan Region (Germany) through a pragmatic trial.[27] If proven effective, the final phase (January to December 2024) will culminate in a plan for expanding the PARS to other regions and integrating it into the broader healthcare system.

Across all phases of the project, we apply a coproduction approach to incorporate expertise from all relevant stakeholders and to facilitate long-term implementation in routine care. Coproduction, a participatory research approach, lacks a universally accepted definition and is variably applied. As per Smith and colleagues, our interpretation of coproduction aligns with the typology of *integrated knowledge translation*, defined as 'a collaborative process in which academic researchers work with 'knowledge users' (eg, clinicians, policy makers, health system leaders, industry partners) in all parts of the research process, from shaping the research question

to implementing the research findings, with the aim of making research more impactful'.[14] A more detailed description of the coproduction method employed in the *BewegtVersorgt* project is presented in the following sections and in a previously published paper.[25]

The qualitative study reported here focuses specifically on the first project phase and uses semi-structured interviews to evaluate stakeholders' experiences of participating in the coproduction process. The evaluation is guided by quality parameters in health promotion and thereby addresses the topics of assessment quality, structural quality and process quality during the joint development phase of the PARS.[28] The following sections provide an overview of the coproduction process for PARS development and the methods of this qualitative study.

## Public and patient involvement

By using a coproduction approach, we included the diverse interests of patients, physicians, health insurance companies, physiotherapists and exercise therapists, and sports clubs. Patients' perspectives were represented throughout the study by three patient organisations: the diabetes association (Deutsche Diabetes-Hilfe-Menschen mit Diabetes e.V.), the rheumatoid arthritis association (Deutsche Rheuma-Liga Landesverband Bayern e.V.) and the centre for patient education and health promotion (Zentrum Patientenschulung und Gesundheitsförderung e.V.). All stakeholders were involved in the design of the PARS, the development of the implementation plan and the evaluation plan, and they will also participate in the dissemination of the results and the development of a transfer and scaling-up concept.

## Process of coproduction

Table 1 gives a chronological overview of the process of coproducing the PARS between June 2019 and December 2020. The process included preparatory activities, four joint coproduction meetings and several bilateral and multilateral meetings.

The preparatory activities, carried out mainly by the research team, aimed to lay the necessary foundations for the joint development of the PARS. Before starting the actual elaboration of the PARS, we thought it was important to gather information on three subjects: (a) the stakeholders' vision of a potential PARS; (b) the identification and analysis of existing healthcare services in the German healthcare system, which addresses PA promotion and (c) the existing international PARS.

The four coproduction meetings, organised and moderated by the research team, were dedicated to the collaborative development of a new PARS model involving twelve organisations (stakeholders or practitioners) from the primary care setting. A detailed description of the coproduction team is given in the study protocol.[22]

A. Two major health insurance companies with three to four representatives per meeting (six representatives in total).

B. Three physician associations with three to four representatives (general practitioners, sports physicians and medical specialists) per meeting (six representatives in total).

C. Four PA professionals' associations with four to five representatives (physical therapists, exercise therapists and exercise instructors) per meeting (nine representatives in total).

D. Three patient organisations with one to three representatives per meeting (four representatives in total).

E. Research team with six to seven representatives per meeting (eight representatives in total).

In between coproduction meetings, the research team organised bilateral or multilateral sessions with different stakeholders, depending on the specific topics that needed careful attention. During these sessions, we delved more deeply into the expertise of each stakeholder. Since the health insurance companies covered the costs of the PARS during the study period based on the German Social Code V (§§ 63 Abs. 2, 64 SGB V), it was imperative to address various legal issues and finalise contracts. This complex process, spanning several months, involved numerous meetings between health insurance companies, physician associations, PA professionals' associations and the research team. Ensuring the intervention's content and the practical feasibility of the referral process required extensive discussions and validation with representatives of scheme participants and deliverers, namely, patient organisations, physician associations and PA professionals' associations. The latter two were also intimately involved in refining the PARS implementation plan.

## Sampling and participants

Based on all stakeholder representatives participating in the four coproduction meetings (n=25), seven interview partners were purposefully selected using maximum variation sampling.[29] Based on the research question, we aimed to capture the most comprehensive picture representing as many facets as possible from stakeholders' experiences with the process. Thus, we decided to interview partners who represent a variety of participating stakeholders regarding (a) the group or sector they or their organisation belongs to (physician associations, PA professionals' associations, patient organisations, health insurance companies), (b) the stakeholders' position within their organisation (management level or non-management level) and (c) the stakeholders' attendance during the development phase (total number of meetings attended). With regard to attendance, we decided that potential interview partners must have participated in at least two meetings to be able to share experiences with the coproduction process.

The characteristics of the final sample (n=7) are presented in table 2. Due to staff restructuring in two patient organisations, only one representative of patients' perspectives was available for an interview. A representative from a physician association could only provide

**Table 1**  Overview of the PARS coproduction process from June 2019 to December 2020

| Coproduction process | Participants | Methods | Results |
|---|---|---|---|
| **June–September 2019** | | | |
| **Individual meetings** with each stakeholder organisation to capture their perception of a PARS | A, B, C, D, E | Semistructured interviews (60 min) | Summary of stakeholders' attitudes and ideas of a PARS |
| **June–October 2019** | | | |
| **Analysis** of existing healthcare services for PA promotion within the German healthcare system | E | Literature search, document analysis | Evidence for the need for a PARS in the German healthcare system |
| **Analysis** of international PARS | E | Literature search | Identification of potential PARS designs and components based on international evidence |
| **October 2019** | | | |
| **First coproduction meeting**: Provision of theoretical foundations for the design of PARS | A, B, C, D, E | Presentations, group work, plenary discussion, knowledge transfer | The need for a PARS in Germany is confirmed by all stakeholder organisations; stakeholders acquired basic knowledge to design PARS |
| **November 2019** | | | |
| **Second coproduction meeting**: Joint development of PARS | A, B, C, D, E | Presentations, knowledge transfer, group work, reflection, plenary discussion | Three potential PARS models were developed |
| **November 2019–January 2020** | | | |
| **Bilateral meetings**: Clarification of financial and legal conditions for implementation of a PARS | A, E | Discussion, negotiation | Preliminary agreement with each healthcare insurance company on financial resources and legal conditions |
| **January 2020** | | | |
| **Third coproduction meeting**: Selection of one PARS model | A, B, C, D, E | Presentations, group work, plenary discussion | Preliminary decision on one PARS model |
| **May–July 2020** | | | |
| **Multilateral meetings**: Clarification of PARS details and costs | A, E | Presentations, discussions | Agreement on costs for PARS delivery to be covered by the healthcare insurance companies |
| **Multilateral meetings**: Refinement of PARS content and clarification of the scheme deliverers' role | B, C, D, E | Presentations, discussions | Agreement on PARS content and scheme deliverers' role |
| **May–November 2020** | | | |
| **Bilateral meetings**: Refinement of PARS implementation plan | B, C, E | Discussions | Agreement on implementation plan, roles and responsibilities |
| **December 2020** | | | |
| **Fourth coproduction meeting**: Consensual approval of the final PARS and implementation plan | A, B, C, D, E | Presentation, group work, plenary discussion | Official agreement on the final PARS and the preliminary implementation plan and trial design |

A, healthcare insurance companies; B, physician associations; C, physical activity professionals' associations; D, patient organisations; E, research team; PA, physical activity; PARS, physical activity referral scheme.

written answers to interview questions due to regulations of the organisation. Thus, we decided to interview another physician representative. Apart from the patient representation, two representatives per sector were interviewed, differing in attendance at coproduction or bilateral meetings (ranging from 4 to 12 meetings) and position within their organisation (management level, n=5; non-management level, n=2). All interviewees were contacted via email and gave their written informed consent for the interview to be conducted, digitally recorded, transcribed and used for the study. The participants were informed about the background and the aim of the study, but details about the researchers' personal interests were not disclosed.

**Table 2** Sample characteristics

| Interview | Organisation | Gender | Attendance* | Position within the organisation |
|---|---|---|---|---|
| 1 | Physician association | Male | 6 | Management level |
| 2 | Physician association | Male | 8 | Non-management level |
| 3 | PA professionals' association | Male | 5 | Non-management level |
| 4 | PA professionals' association | Female | 4 | Management level |
| 5 | Patient organisation | Male | 4 | Management level |
| 6 | Health insurance company | Female | 12 | Management level |
| 7 | Health insurance company | Male | 5 | Management level |

*Total number of meetings (coproduction meetings and bilateral meetings) attended during the development phase.
PA, physical activity.

## Data collection

All interviews were conducted as individual sessions through video calls (n=6) or face-to-face interviews (n=1) between June and August 2022. The duration of the interviews ranged from 45 to 70 min. We digitally recorded the interviews, a trained student assistant transcribed the audio records verbatim following a transcription guideline,[30] and text passages used for publication were translated into the English language. We extended an invitation to the interviewees to review their transcripts, but none opted to take advantage of this opportunity.

According to the exploratory character of the interviews, we developed semi-structured interview guidelines (see online supplemental file 1) based on literature on quality parameters in health promotion and on previous projects using cooperative planning for PA promotion.[28 31 32] The interview guide was also discussed with the entire research team that accompanied the development process and pilot-tested with one stakeholder who was not part of the final interview sample. The main focus of the interview guide was on stakeholder experiences with the coproduction process, particularly on the appropriateness of the participatory approach, the roles of the different stakeholders and the barriers and facilitators they met to contribute to the development of the new PARS. To enhance the interview flow and obtain more detailed information, specific questions were posed, addressing aspects such as group composition, collaboration, participation or potential process adaptations. In addition, individual questions were integrated as needed when interviewees raised new and unexpected topics.

Interviews were led by SK (female, PhD, research associate), who got to know all stakeholders during the development process and worked towards establishing a trusting relationship with the interviewees. Data analyses were conducted by SK in close consultation with IN (female, MA, PhD student); both have expertise in qualitative research methods and work as researchers in the field of PA promotion.

## Data analysis

We applied summarising qualitative content analysis to analyse the interview material.[33] This analysis method is similar to inductive coding but a comparatively extensive procedure that aims to reduce the material to the essential content related to the research question. All interview data were managed (transcription and analysis) using MAXQDA 2022 (VERBI GmbH).

First, all transcripts were read several times, and interesting text passages were highlighted in order to become familiar with the interview material. Throughout the analysis, we followed the step-by-step model of summarising content analysis and applied the corresponding interpretation rules (paraphrasing, generalisation, reduction of the material).[33] The result of this reduction process was a preliminary category system summarising stakeholder experiences. The category set was re-tested on the original interview material by SK and IN (exemplary three transcripts), which discussed disagreements and adapted the categories accordingly. During re-testing, SK and IN revised the coding scheme by allocating the categories to facilitators, challenges and suggestions for process adaptations in order to enhance the interpretation of data. SK applied the final category system to all seven interviews and discussed unclear text passages with IN.

## RESULTS

### Overview of findings

The final category system comprised 294 codings in total, with four themes that were relevant to reflect stakeholder experiences with the coproduction process: method and structure of the process; group composition and level of participation; collaboration between stakeholders; and the role of researchers. Table 3 provides an overview of the identified themes and the associated facilitators and challenges. Most codings (164 codings) were attributed to facilitators, and a few statements (98 codings) were associated with challenges. Suggestions for adaptations of the development process were identified in two themes (group composition and participation; collaboration between stakeholders), with 32 coded segments in total.

### Facilitators

The most frequently mentioned facilitator of the development process was the involvement of stakeholders using

**Table 3** Facilitators, challenges and adaptations of the development process from a stakeholder perspective

| Themes | Facilitators (categories) | Frequency Segments*/Interviews† | Challenges (categories) | Frequency Segments*/Interviews† | Adaptations (categories) | Frequency Segments*/Interviews† |
|---|---|---|---|---|---|---|
| Method and structure of the process | | | | | | |
| | Coproduction approach | 30/7 | Merging of different perspectives | 8/5 | – | |
| | Process of coproduction | 19/6 | Structure and content of meetings | 9/1 | | |
| | Structure and content of meetings | 21/4 | | | | |
| Group composition and level of participation | | | | | | |
| | Multi-sector stakeholder group | 14/6 | Roles and hierarchy | 31/7 | Integration of other stakeholder groups | 13/4 |
| | Possibility of active participation | 19/6 | Engagement of all stakeholders | 11/3 | Integration of local actors | 8/3 |
| | | | | | Appropriate stakeholder participation | 5/2 |
| Collaboration between stakeholders | | | | | | |
| | Communication, atmosphere, interaction | 29/7 | Clarification of intervention costs | 11/4 | Increase knowledge exchange | 4/2 |
| | Sharing knowledge and creating mutual understanding | 12/4 | Competition and conflicting interests | 9/4 | Break up competition | 2/1 |
| | | | Knowledge gaps and lack of understanding | 10/2 | | |
| Role of researchers | | | | | | |
| | Coordinating role | 20/6 | Steering role | 9/1 | – | |
| | **Total number of coded segments** | **164** | | **98** | | **32** |

*Total number of coded segments per category.
†Total number of interviews in which category was identified.

a coproduction approach (*coproduction approach*). The interviewees argued that the participation of the stakeholder organisations was an important success factor, not only for the development of a PARS but also for accelerated joint decision-making for one PARS model. Stakeholders were able to express their needs and contribute their expertise, which improved their sense of ownership.

> I am very sure that, if we had not been involved in the process, but someone had presented me exactly the same PARS as a finished thing, and if I had not seen how the whole thing was developed and how it was discussed, then I think it would have failed, if I may say so here. (A representative of a physician association)

Overall, the stakeholders described the development process as being goal-oriented, constructive, inspiring and well scheduled (*process of coproduction*). They also assessed the duration of the process and the number of meetings as appropriate. Furthermore, the three joint coproduction meetings at the beginning of the development phase, which lasted 5–6 hours, seemed to be very important (*structure and content of meetings*). The interviewees argued that this enabled them to build relationships with each other and deepen the knowledge exchange. The coproduction meetings contained a good mixture of knowledge transfer, group work and plenary discussions. Some of the stakeholders pointed out that the interdisciplinary group work (3–5 people per group) in particular had contributed decisively to the development of the PARS by giving them room for creativity and intensive discussions.

> I think it helped a lot that you did these planning rounds, formed working groups for the corresponding packages and mixed the stakeholders. Because I think, that the vision of the individual stakeholders comes to life when you built working groups like that. … what I believe is that these working groups and the mixture of the single stakeholders have practically untied the knot there. (A representative of a health insurance company)

All interviewees perceived the overall cooperation between stakeholders as constructive and positive. Particularly frequently emphasised were the open communication, the respectful and collegial interaction with each other and the good atmosphere at the joint coproduction meetings (*communication, atmosphere, interaction*). Some of the stakeholders also mentioned that sharing knowledge with each other was very interesting and important to get a mutual understanding of the different perspectives and problems of other organisations. This has contributed to a common understanding and led to joint decision-making (*sharing knowledge and creating mutual understanding*).

> That is always an enrichment as well, because you also take something out of this and say: 'Aha'. You gain an understanding of others. But also the others

perhaps show understanding towards the health insurers, who say: 'Well, we'd like to but the legal basis simply doesn't allow it'.(A representative of a health insurance company)

Most interviewees reported that they felt actively engaged in the process and that all stakeholders had the opportunity to talk about their needs and ideas (*possibility of active participation*). Different opinions were heard, accepted and embraced. In almost all interviews, the group composition was described as very fitting, with all relevant stakeholders represented (*multi-sector stakeholder group*). A specialty of the consortium was the large number of organisations from different sectors across the healthcare system. Although most of the participants reported that it was the first time they had worked in such a diverse and large team, they perceived this as a very positive experience.

> The selection was very good because it's a good mix of science, funding and implementation. So, that was absolutely very good. (A representative of a PA professionals' association)

Almost all interviewees mentioned the research team as a facilitator of the process (*coordinating role*). They perceived the researchers as coordinators and moderators by describing their role as supporting and accompanying. Some of the stakeholders argued that the research team provided a kind of interface where information was collected and spread out, which was seen as an important structure for process coordination.

> I would definitely say that was a guarantee of success for the methodology that you led there in the right dose. (A representative of a PA professionals' association)
>
> I: How did you feel about how the participation process was controlled by us to a certain extent? A2: Yes, I thought it was totally good. I think it was important. I think it's good to have a strong organization overseeing and moderating the whole process, so that it doesn't get talked up or degenerate into some kind of trench warfare … I thought that was good. Otherwise, I don't think it would have worked so well. (A representative of a physician association)

### Challenges
In terms of group composition and level of participation, it was clear across all interviews that there were differences in hierarchy and role allocation between participants (*roles and hierarchy*). The stakeholders had the impression that the health insurance companies had the strongest impact on the development of the PARS since they were the funders and had to embed the intervention within the legal requirements. Representatives of physician associations and PA professionals' associations were also seen as playing a central role in the coproduction process.

I had the impression, of course, that health insurance representatives play a very important role in any case. Without them, nothing works. Everything stands and falls with it. I think those were the three main players, right? The health insurance representatives, of course, the family doctors, and then the service providers on site. (A representative of a physician association)

In comparison, patient organisations have taken a rather inconspicuous role, which changed in the course of the process. Furthermore, some of the stakeholders playing a central role expressed that while they generally felt actively involved in the process, they would have liked to be even more involved and have more influence (*engagement of all stakeholders*).

Interviewer: How would you describe your role in this development process? Representative of a patient organisation: My role? First of all, as a listener. … That was important for me to know, first of all, who actually belongs in which field. What dependencies already exist? (A representative of a patient organisation)

In some interviews, it turned out that there were also challenges in the collaboration between stakeholders. Some argued that many discussions focused on intervention costs, thereby pushing the development of interventional components of the PARS into the background (*clarification of intervention costs*). Many ideas have failed because of health economic efficiency concerns.

Because there is a pot full of money, and there is no more money. And how you distribute that is always a 'Herculean task'. (A representative of a health insurance company)

Another challenge that became apparent during the collaboration was the competitive situation between individual organisations (*competition and conflicting interests*). Some interviewees pointed out that cooperation between representatives was limited because some organisations have different goals that compete with each other. In particular, stakeholders from PA professionals' associations noted that some representatives were more engaged in lobbying, potentially limiting open and constructive collaboration. Furthermore, two interviewees commented that cooperation was hampered by a lack of knowledge and understanding among stakeholders (*knowledge gaps and lack of understanding*). For some representatives, it was difficult to leave their own position and to take a different perspective during discussions of complex problems. This has influenced the cooperation and slowed down the joint decision-making process.

I think everyone is always surprised by the others about what they do and where the problem areas lie. Yes, and also sometimes, probably also a certain: 'I can't understand that at all! Why don't we just do that now?' (A representative of a health insurance company)

In the majority of interviews, it was pointed out that the involvement of stakeholders using a coproduction approach in some way was also a challenge to the development process because so many different perspectives had to be merged (*merging of different perspectives*). The involvement of a large number of stakeholder organisations presented the challenge of incorporating a multitude of diverse opinions into the process, requiring careful consideration. Finding compromises and solutions that satisfied everyone proved to be challenging and time-consuming. One interviewee also commented that the coproduction process was too freely designed at the beginning and was therefore less goal-oriented and efficient (*structure and content of meetings*).

… of course, one must not forget that there are worlds colliding, right? And synchronizing all that is just a bit difficult. (A representative of a health insurance company)

With regard to the researchers' role, one stakeholder perceived the research team as too steering and controlling the development process (*steering role*). He criticised the central role of the researchers because this had restricted stakeholder participation in his opinion. The interviewee argued that, while it is understandable that such processes require a certain stringency, the research team brought their own perspective too much into the process.

Also, the co-production meetings with group work, which were also very well prepared, is a fine line between preparing very well and steering. So I have always felt that there was little room for creativity or leaving the path and going into a completely different path. (A representative of a PA professionals' association)

### Suggested adaptations

Overall, only a few ideas for adjustments to the process emerged from the interviews. Some stakeholders reported that it could be beneficial to integrate even more organisations into such a coproduction process, for example, cross-cutting patient organisations that represent the needs of diverse patient populations (not just those with a specific condition), organisations responsible for the billing of medical services, further health insurance companies and the German pension insurance company. Moreover, some interviewees pointed out that the group should be extended by integrating more local actors (eg, local sports clubs and exercise providers), which would support the development of a local network. Furthermore, two interviewees wanted even more stakeholder involvement in the process. Recognising that ideas of appropriate participation can vary significantly, some stakeholders proposed the approach of asking in advance about the extent to which stakeholders want to be involved.

In order to increase the collaboration between participants, some have called for more knowledge exchange between the different stakeholder groups. Each representative should be given the opportunity to make a presentation where they can provide expert knowledge on a particular topic from their perspective. In particular, with regard to the healthcare system and its legal conditions for implementing new health services, the interviewees wanted to have more knowledge transfer. To further improve collaboration in such a process, one interviewee suggested addressing competitive thoughts between individual associations in advance. However, he also pointed out that it could be difficult to address such a sensitive topic, and it is questionable whether this will succeed.

## DISCUSSION

Despite the perceived potential of coproduction approaches in developing and implementing new health services, there is limited evidence of their application in the field of PARS research. This qualitative study points out both the facilitators and challenges, as well as potential adaptations, of a joint development of a PARS within the German healthcare system. Overall, stakeholder experiences with the development process were overwhelmingly positive, mainly highlighting facilitators, identifying some challenges and proposing a few suggestions for adaptations. All interviewees pointed out that the coproduction approach and its methodical implementation were paramount facilitators for PARS development. In contrast, differences in roles and hierarchies among stakeholders appeared to be the primary challenge, potentially disrupting the process.

Few studies have reported on coproduced PARS, and even fewer have evaluated the joint development process.[16 17 34–36] The *Co-PARS* project serves as one example, sharing many similarities with our *BewegtVersorgt* project in terms of methodical approach, process evaluation and findings.[17 35] In that study, facilitators of the process included multidisciplinary perspectives, working in subgroups and having multidisciplinary debates. This aligns with our findings, where stakeholders valued the integration of diverse actors from the German healthcare sector, especially highlighting the importance of small working groups for discussing various perspectives. Furthermore, Buckley and colleagues identified contrasting views and power imbalances among stakeholders as challenges. This is similar to the statements from our interviews and aligns with broader literature that recognises these challenges and barriers as typical in coproducing research in health science.[14 37]

In comparing our study with the *Co-PARS* project, a notable difference lies in the composition of the planning groups and the approach to handling differences in roles and hierarchy. Buckley and colleagues specifically focused on the integration of patients in the coproduction process and employed different treatment strategies for various stakeholder groups (eg, separating staff from managers, holding separate meetings with service users) to ensure equal representation.[17 35] In contrast, while we also included patient representatives, they did not advocate for individual needs as service users but rather presented the perspective of their respective patient organisation. Since we used a coproduction approach based on the principles of *integrated knowledge translation*, we did not primarily aim at altering power structures among stakeholders[14] but to create opportunities for all representatives to build relationships and actively participate in planning meetings. The evaluation of our approach revealed high satisfaction among most stakeholders, particularly patient representatives, indicating a sense of ownership throughout the process. This collaborative effort successfully brought stakeholders closer together, achieving the objective of the first project phase—a joint decision on a PARS model that garnered support from all involved parties.

### Implications for practice and policy

Coproduction within healthcare systems inevitably involves natural power differentials among various actors.[37] In the *BewegtVersorgt* project, stakeholders perceived health insurance representatives as crucial partners in the development process because of their position in the German healthcare system, their knowledge of the system and their role as intervention funders. On the one hand, they were great supporters in the process because they were actively engaged in shaping the content of the PARS. The PARS can be tested in routine care following the regulations of the German Social Code. On the other hand, the issue of financing the intervention was considered by some stakeholders to be very conflictual and to have hindered the joint development process. Furthermore, it also became evident that health insurance representatives possessed specific and selective knowledge in their care sector due to the complexity of the system. Even for them, as representatives of the system, grasping all possibilities at a higher level and linking and using information posed challenges. However, in the *BewegtVersorgt* project, understanding the structures and processes of the German healthcare system was inherently important because the newly developed PARS needed to align with the existing system—an observation noted in other studies as well.[17] From our perspective, integrating health insurance companies into the coproduction of PA-related health services is indispensable, but it is important to be aware of their specific roles and positions in the process and the challenges of implementing innovations in rigid healthcare systems. Therefore, researchers must possess a thorough understanding of existing structures and potential connections within the complex healthcare system (as illustrated in 'preparatory activities', see table 1). This knowledge empowers researchers to facilitate the integration of innovative interventions into the system. The moderating role of the research team, encompassing not only PARS but also the coproduction and implementation of other innovative healthcare services, remains vital.

Given the potential power imbalances among stakeholders, we believe it is crucial for a research team to be aware of these dynamics and make informed decisions on how to address such challenges within a coproduction process. Smith *et al* describe the process of sharing power in coproduction as rather fluid and fragile, which requires careful negotiation.[14] Buckley *et al* point out that solving the challenges of coproduction 'requires leadership, a tolerance of messiness, and careful negotiation of group politics (particularly when the group involves natural power imbalances, eg, commissioners and service providers)'.[35] In the *BewegtVersorgt* project, we have taken on such a coordinating and guiding role as well, and we constantly had to weigh out how much 'guidance' the group needed in order to achieve a shared decision. Due to the above-mentioned central role of health insurance companies, they were involved in most of the bilateral meetings with other stakeholder organisations, and there was intense communication with us researchers during the development process. Therefore, the challenge for us was to ensure that the other partners had enough room to actively participate in the process, to bring in their expertise and to be sufficiently involved in the decision-making process. Researchers appear to play a critical role in this moderating process and should bring a wide range of skills and capacities (eg, good communication skills, ability to manage conflicts, being versatile and adaptable to different situations).[37] However, a clear guideline on how scientists can best support the joint development and decision-making process seems difficult to realise due to the individuality and complexity of such coproduction processes. One interviewee, who was critical of the researchers' role as too steering, suggested that stakeholders should be asked up front how they would like to see participation and that a common understanding should be reached. Openly addressing the different roles of actors and potential power differences may assist in determining how to deal with these imbalances and the extent to which researchers should engage in negotiation.

The way researchers manage various roles and power imbalances among stakeholders in joint development processes is intricately linked to the chosen coproduction method. Smith and colleagues essentially distinguish between three types of coproduction: *citizens' contributions to public services* (type 1), *integrated knowledge translation* (type 2) and *equitable and experientially informed research* (type 3).[14] While type 1 focuses on improving and understanding public services, types 2 and 3 directly translate to the research context. *Integrated knowledge translation* and *equitable and experientially informed research* both strive for shared decision-making, trustful relationships and open communication between partners. Type 2, primarily developed by health researchers and funders, targets the translation of knowledge into practice, while type 3 includes approaches such as participatory action research or community-led research, which focus on equality and empowerment of disadvantaged people. Consequently, *equitable, and experientially informed research* aims to break

down traditional hierarchies of power, and people with lived experiences (eg, service users, patients) are essential partners of the whole process. In contrast, *integrated knowledge translation* does not primarily focus on altering power structures between stakeholders and patients, or service users are not required for the research process. In developing the PARS in the *BewegtVersorgt* project, we were strongly guided by the concept of cooperative planning, which, according to Smith's typology, follows the principles of *equitable and experientially informed research*.[14 38] In particular, we adopted the process structure, planning meetings, methodical design and consensus-based decision-making from this concept. However, we had to modify the concept mainly due to the key role of health insurance companies by accepting power differences between stakeholders to a certain extent.[25] Furthermore, as the goal of the coproduction process—the development of a PARS—was clear from the outset, we, as the research team, contributed much of our knowledge to the process at the beginning to ensure that all stakeholders had a similar level of knowledge. If we now take into account the adaptations we have made to cooperative planning, especially regarding managing power differences, the approach in the *BewegtVersorgt* project corresponds most closely to *integrated knowledge translation* in our understanding. Based on our experience, we would recommend using coproduction for the development of PARS, especially for countries whose healthcare systems have a similar structure to the German system. Nevertheless, this study also provides relevant information for using coproduction processes to develop and implement innovative forms of care in other countries and their healthcare systems. Further, given the vast variety of coproduction approaches, we support the idea of distinguishing between different types and thus selecting and potentially adapting the appropriate method based on the project objectives.[14]

## Limitations

Some limitations should be taken into consideration when interpreting the findings of this study. First, the development process, to which the interviews refer, was conducted between June 2019 and December 2020. However, interviews with stakeholders could only be conducted between June and August 2022 due to other priorities in the implementation process, so there may be memory bias. Second, the intended variability of the interview sample may be limited because only one patient representative was available for interviews (compared with two representatives from all other sectors), and more stakeholders from the management level than the non-management level were interviewed due to an uneven distribution in the planning group. Third, the interviewer (SK) was involved in the coordination and moderation of the process together with other project members, so social desirability response bias cannot be ruled out. Finally, this study does not represent a complete evaluation of the coproduction process, but is limited to the subjective experiences of the stakeholders.

## Conclusions

Coproduction is a useful method for the development of PARS intended for integration into healthcare systems. When considering the typical challenges of coproduction processes, such as dealing with natural power differences among healthcare system actors, a careful decision about the appropriate type of coproduction for the PARS development becomes crucial. The research team plays a central role in such joint development processes and should be equipped with a diverse set of skills and capacities. However, there is currently a lack of clear guidance on how the research team can best support the consensus-finding process. To further enhance the evidence base for the coproduction of PARS, future studies need to report on and evaluate the development processes of new PARS models more frequently and systematically.

**Acknowledgements** We would like to thank the following partners for their cooperation in the BewegtVersorgt project: Bayerischer Landesärztekammer (BLÄK), Bayerischer Hausärzteverband (BHÄV), Bayerischer Sportärzteverband (BSÄV), Deutscher Verband für Gesundheitssport und Sporttherapie (DVGS), Bundesverband selbstständiger Physiotherapeuten (IFK e.V), VDB-Physiotherapieverband, Deutscher Olympischer Sportbund (DOSB), Deutsche Diabetes-Hilfe-Menschen mit Diabetes e.V. (DDH-M), Deutsche Rheuma-Liga Landesverband Bayern e.V., AOK Bayern – Die Gesundheitskasse, DAK-Gesundheit (Landesverband Bayern), and Zentrum Patientenschulung und Gesundheitsförderung (ZePG e.V.). We would also like to thank all the interviewed representatives for sharing their experiences with us.

**Contributors** KP conceived the initial idea for this project and along with WG, KA-O and PG, submitted the funding application for the study. SK and EM designed this qualitative study. SK conducted the interviews and analysed the data together with IN. SK and EM drafted the manuscript. AW, WG, PG, KA-O and KP contributed to the study design and provided feedback on the manuscript. All authors reviewed and approved the final manuscript. SK is responsible for the overall content as guarantor.

**Funding** This work was supported by the Federal Ministry of Health based on a resolution of the German 'Bundestag' by the federal government [grant number: ZMV I 1 – 2519FSB109].

**Competing interests** None declared.

**Patient and public involvement** Patients and/or the public were involved in the design, or conduct, or reporting, or dissemination plans of this research. Refer to the Methods section for further details.

**Patient consent for publication** Not applicable.

**Ethics approval** This study involves human participants and was approved by ethics committee of the Friedrich-Alexander-Universität Erlangen-Nürnberg (ethics approval number: 331_20 B). Participants gave informed consent to participate in the study before taking part.

**Provenance and peer review** Not commissioned; externally peer reviewed.

**Data availability statement** Data are available upon reasonable request. Data are available upon reasonable request from the corresponding author.

**ORCID iDs**
Sarah Klamroth http://orcid.org/0000-0002-1302-1842
Eriselda Mino http://orcid.org/0000-0002-1885-0009
Anja Weissenfels http://orcid.org/0000-0002-3271-4935

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
