## [Reviewer comments · BMJ Open]

ARTICLE DETAILS

TITLE (PROVISIONAL)	Co-producing a physical activity referral scheme in Germany: A qualitative analysis of stakeholder experiences
AUTHORS	Klamroth, Sarah; Mino, Eriselda; Naber, Inga; Weissenfels, Anja; Geidl, Wolfgang; Gelius, Peter; Abu-Omar, Karim; Pfeifer, Klaus

VERSION 1 – REVIEW

REVIEWER	Searle, Aidan National Institute for Health Research (NIHR) Biomedical Research Unit in Nutrition, Diet and Lifestyle at the University Hospitals Bristol NHS Foundation Trust and the University of Bristol, Bristol Biomedical Research Centre
REVIEW RETURNED	05-Jan-2024

GENERAL COMMENTS	This qualitative study investigated stakeholders' experiences with the development of a physical activity referral scheme (PARS) in the German healthcare system using a co-production approach from the viewpoint of stakeholders in this process. It is a well conducted study, well reported and informative. However, I have a few comments and suggestions to make before recommending publication in BMJ Open. Firstly, as this work was conducted in 209/2020 what developments have been made in implementing PARS in the German healthcare system - what, if any, were there pragmatic outcomes in undertaking this research? Sampling With regard to sampling the interviewee's from the co-development meetings it is unclear how this was achieved chronologically if the first stakeholder interviews were in June-September 2019 and the first co-production meeting was in October 2019 - can this be clarified? As the number of interview participant is small n =7 did the researchers consider additional online consultation methods with a broader base of stakeholders to enhance of confirm the facilitators and challenges identified from these interviews as part of the co-development process? Methods How were the topics chosen to structure the interviews - were they informed by the literature search or plenary discussions referred to in Table 1? Discussion
---

	I am interested in how the findings can be transposed or relate to the UK National Health Service (NHS) - what learnings are there for potential private intervention in the NHS as this aspect may be pertinent to readers of this journal? Also, with regard to reflexivity of the interviewer / researchers and the comment from one of the stakeholder representatives (from PA organisation?) such that the research team were felt to be too steering in the co-production process - could this point be expanded upon further as a potential limitation?
--	---

REVIEWER	Spiteri, Karl Government of Malta Ministry for the Family and Social Solidarity
REVIEW RETURNED	16-Jan-2024

GENERAL COMMENTS	Thank you for asking me to review this article below are my comments: Abstract The objective of the study could include the term 'evaluation' of the process to make it clearer to the reader as at first it is unclear. Include the methodological approach adopted within the qualitative study. The focus in the participants should be on the seven stakeholders not on the group developing the PARS as the study is about the data gather from this group. Introduction The objective of this research needs to be made more explicit in the end of the chapter. More information on process evaluation needs to be added in the introduction as the focus of the study is the evaluation process of the PARS development. Method This section focuses on the actual project and information about how the project took place which is required. However, there needs to be more focus on how the methodological approach which guided the evaluation process. There needs to be a clearer distinction between the process being evaluated and the current study. Results: The results section is well presented and succinct. Can suggestions be included as one of the themes and the suggestion grouped whether it was an issue with method, group composition ect. So, it can be included in table 3. Discussion The issue of power struggle is very interesting in this co-production process. However, possibly it can be given further value within the discussion if this is formulated within the methodological perspective of the study. The perspective of how people create knowledge has an impact on the interpretation of power differentials.
---

VERSION 1 – AUTHOR RESPONSE

Comments Reviewer 1

This qualitative study investigated stakeholders' experiences with the development of a physical activity referral scheme (PARS) in the German healthcare system using a co-production approach from the

viewpoint of stakeholders in this process. It is a well conducted study, well reported and informative. However, I have a few comments and suggestions to make before recommending publication in BMJ Open.

Response: Thank you very much for reviewing this manuscript and your interest in our study and the BewegtVersorgt project.

Below you can find the point-by-point response to your comments.

1) Firstly, as this work was conducted in 2019/2020 what developments have been made in implementing PARS in the German healthcare system - what, if any, were there pragmatic outcomes in undertaking this research?

Response: For the implementation and evaluation of the co-produced PARS (project phase 3), we have conducted a pragmatic trial in the Nuremberg Metropolitan region for a period of two years (December 2021 to December 2023). An overview of the project phases is provided in the first paragraph of the methods section. For a better understanding, we have added the time intervals of the individual phases

(methods, section 'study design', page 6, lines 139-150). The pragmatic trial was conducted in a hybrid II-design, in order to evaluate both the effectiveness and the implementation success of the PARS. We

are currently analyzing the data and will publish the results as soon as possible. Further details about the pragmatic trial can also be found in the study protocol: Weissenfels A, Klamroth S, Carl J, et al. Effectiveness and implementation success of a co-produced physical activity referral scheme in Germany: study protocol of a pragmatic cluster randomised trial. BMC Public Health 2022;22(1):1545. doi:10.1186/s12889-022-13833-2.

2) Sampling

With regard to sampling the interviewee's from the co-development meetings it is unclear how this was achieved chronologically if the first stakeholder interviews were in June-September 2019 and the first co-production meeting was in October 2019 - can this be clarified?

Response: Thank you very much for this comment. Probably the term "individual stakeholder interviews"

(June – September 2019) in table 1 is confusing and has led to a mix-up with the interviews of the qualitative study. These are two different things and to avoid further confusion, we have replaced the term "individual stakeholder interviews" in table 1 with the term „Individual meetings with each stakeholder organisation“ (see table 1, page 9). We have conducted these individual meetings at the beginning of the process to gather information on stakeholders' vision of a potential PARS. These individual stakeholder meetings (June - September 2019) were not related to the stakeholder interviews

we conducted between June and August 2022 to evaluate the co-production process in this qualitative

study (methods, section paragraph 'data collection', page 11, lines 245-246).

3) As the number of interview participant is small n =7 did the researchers consider additional online consultation methods with a broader base of stakeholders to enhance or confirm the facilitators and challenges identified from these interviews as part of the co-development process?

Response: Thank you, this is a very important comment. We agree that the sample appears to be rather

small, and that integrating the perspectives of all stakeholders participating in the process would have strengthened the results. However, stakeholders were purposefully selected for interviews "to capture the most comprehensive picture representing as many facets as possible from stakeholders' experiences". Based on the principles of qualitative sampling, we have selected information-rich cases

that differ on specific criteria and thus reflect the broadest possible range of experiences (methods, section 'sampling and participants', page 10, lines 221-230).

Nevertheless, we had considered options such as a short survey of all stakeholder to confirm or enhance the results from the interviews. There were two reasons why we decided against this, both related to the large time gap between the development of the co-production process and its' evaluation. First, there is the problem of memory bias. We already have addressed the memory bias as a limitation for the interviews in the manuscript (page 22, lines 566-569). A survey integrating the interview results would have been conducted at an even later point in time, and the memory bias would have limited the significance of the results. Second, responsibilities in organisations sometimes change quickly and some stakeholders who were involved in the development process were no longer responsible for the BewegtVersorgt project during the second project phase or had left the organisation. The number of stakeholders who could have taken part in such a survey was therefore very small.

4) Methods

How were the topics chosen to structure the interviews - were they informed by the literature search or plenary discussions referred to in Table 1?

Response: Thank you for this valuable comment. The selection of the interview topics was guided by literature on quality parameters in health promotion (in the context of the German healthcare system) and was based on findings in previous projects using cooperative planning for physical activity promotion. We also discussed the topics of the interviews within our research team which has an extensive expertise with cooperative planning processes. We have also integrated this information and

the corresponding references in the methods section of the manuscript (section 'data collection', page 11, lines 252-256).

5) Discussion

I am interested in how the findings can be transposed or relate to the UK National Health Service (NHS) - what learnings are there for potential private intervention in the NHS as this aspect may be pertinent to readers of this journal?

Response: Thank you, this is an interesting thought. One of the reasons why we decided to share our learnings with the co-production approach in the BewegtVersorgt project was to inform researchers and

actors from other healthcare systems. Although co-production processes appear to be highly context specific and need to be adapted to individual needs, we believe that our findings provide relevant

information for the use of co-production processes to develop and implement new forms of care (such as PARS) in other countries, including the UK. In particular, there are some parallels between Germany

and the UK in terms of care structures within the healthcare systems, and in the UK there are many PARS

initiatives to promote physical activity in primary care. In order to highlight the relevance of our findings

for other healthcare systems such as the NHS, we have adapted the section 'Strengths and limitations'

(page 3, lines 70-72) and added some more information in the discussion section (page 22, lines 557-561).

6) Also, with regard to reflexivity of the interviewer / researchers and the comment from one of the stakeholder representatives (from PA organisation?) such that the research team were felt to be too steering in the co-production process - could this point be expanded upon further as a potential limitation?

Response: Thank you for this comment. As reported in the results section, the majority of stakeholders

evaluated the role of researchers as a facilitator of the process and only one stakeholder argued that the researchers may have steered the process too much. During the interview the same person also pointed out that there are different ways to create participation and therefore suggested to create a common understanding of participation up front. To make this more explicit, we have reformulated this sentence in the discussion section (page 21, lines 524-527). From our perspective, the researchers can play a pivotal role in a co-production process and they can guide the process in different directions. Therefore, it is highly relevant that researchers are aware of their role and shape it from the outset in accordance with the chosen co-production method. In the discussion section of the manuscript, we discussed the moderating role of researchers (our role) in managing power imbalances among stakeholders and pointed out how this is intricately linked to the chosen co-production method (discussion, section 'implications for practice', paragraphs 2 & 3). We therefore would refrain from listing the 'steering role of researchers' explicitly as a limitation.

Comments Reviewer 2

Response: Thank you very much for reviewing this manuscript and your valuable comments. Below you can find our point-by-point response to your comments.

1) Abstract

The objective of the study could include the term 'evaluation' of the process to make it clearer to the reader as at first it is unclear. Include the methodological approach adopted within the qualitative study. The focus in the participants should be on the seven stakeholders not on the group developing the PARS as the study is about the data gather from this group.

Response: Thank you for your valuable comment. We have re-arranged the abstract (page 2) in the sections 'objectives' (lines 33-34) and 'participants' (lines 40-44). In order to clarify methodological approach, we made some adaptations in the methods section (please see comment 3).

2) Introduction

The objective of this research needs to be made more explicit in the end of the chapter. More information on process evaluation needs to be added in the introduction as the focus of the study is the evaluation process of the PARS development.

Response: Thank you for this hint. We have reformulated the sub-objectives to better explicate the focus

of our study (Introduction, page 5, lines 132-137). We also added more information about the focus and

the chosen method of the study in the methods section of the manuscript, as we think this information is best placed in the methods section (see response to comment 3).

3) Method

This section focuses on the actual project and information about how the project took place which is required. However, there needs to be more focus on how the methodological approach which guided the evaluation process. There needs to be a clearer distinction between the process being evaluated and the current study.

Response: Thank you for your valuable comment. We have added further information in the methods section (page 6, lines 162-167) to better clarify the focus of the study and its' theoretical basis as well as

the structure of the methods section.

It is important to note that this study does not represent a comprehensive process evaluation of the entire BewegtVersorgt project. As we have clarified in the study objectives and the methods section, this

study only evaluates the first project phase (development of the PARS) and specifically addresses the subjective experiences of stakeholders that have participated in the co-production process. Consequently, this study represents only one part of a comprehensive process evaluation and uses just one specific method (qualitative interviews) rather than a mixed-methods approach. As we conducted the interviews retrospectively, rather than conducting a mixed-methods process evaluation parallel to the PARS development, we have added this as a potential limitation at the end of the discussion section (page 22, lines 574-576). Nevertheless, we believe that our study represents an important contribution to the further development of PARS research, as there are only very few studies to date that focus on the development process.

4) Results:

The results section is well presented and succinct. Can suggestions be included as one of the themes and the suggestion grouped whether it was an issue with method, group composition ect. So, it can be included in table 3.

Response: Thank you for this comment. We appreciate this idea as it gives a good overview and allows for comparison between facilitators, challenges and adaptations within the themes. We adapted table 3 (page 13) and integrated the 'adaptations' accordingly.

5) Discussion

The issue of power struggle is very interesting in this co-production process. However, possibly it can be given further value within the discussion if this is formulated within the methodological perspective of the study. The perspective of how people create knowledge has an impact on the interpretation of power differentials.

Response: Thank you for this relevant hint. For a better guidance of the reader, we have included some further information about the scope of the interviews in the methods section (page 11, lines 257-260). Doing this, we would at the same time refrain from adding further aspects in the discussion as we think that the issue of power differentials is already under comprehensive consideration there (section 'implications for practice', page 19-22).

VERSION 2 – REVIEW

REVIEWER	Searle, Aidan National Institute for Health Research (NIHR) Biomedical Research Unit in Nutrition, Diet and Lifestyle at the University Hospitals Bristol NHS Foundation Trust and the University of Bristol, Bristol Biomedical Research Centre
REVIEW RETURNED	21-Mar-2024
GENERAL COMMENTS	Thanks for the revisions you have made on the basis of the 2 reviews and editors' comments. I feel the article has subsequently improved and is now acceptable for publication in BMJ Open
REVIEWER	Spiteri, Karl Government of Malta Ministry for the Family and Social Solidarity
REVIEW RETURNED	01-Apr-2024
GENERAL COMMENTS	The authors addressed the comments raised appropriately.

VERSION 2 – AUTHOR RESPONSE